# Facile Construction of Polypyrrole Microencapsulated Melamine-Coated Ammonium Polyphosphate to Simultaneously Reduce Flammability and Smoke Release of Epoxy Resin

**DOI:** 10.3390/polym14122375

**Published:** 2022-06-12

**Authors:** Feiyue Wang, Jiahao Liao, Long Yan, Mengtao Cai

**Affiliations:** Institute of Disaster Prevention Science and Safety Technology, School of Civil Engineering, Central South University, Changsha 410075, China; wfyhn@163.com (F.W.); 194812294@csu.edu.cn (J.L.); 194812293@csu.edu.cn (M.C.)

**Keywords:** epoxy resin, ammonium polyphosphate, polypyrrole, flame retardancy, smoke suppression

## Abstract

A unique mono-component intumescent flame retardant, named PPy-MAPP, of which melamine-coated ammonium polyphosphate (MAPP) was microencapsulated by polypyrrole (PPy), was synthesized and carefully characterized. The obtained PPy-MAPP was applied to epoxy resin (EP) for obtaining flame-retarded EP composites. The results show that PPy-MAPP imparts better flame retardancy and smoke suppression properties to EP compared to the same addition of MAPP. The EP composite with 15 wt% PPy-MAPP easily passes the UL94 V-0 rating and achieves an LOI value of 42.4%, accompanied by a 61.9% reduction in total heat release (THR) and a 73.9% reduction in total smoke production (TSP) when compared with pure EP. The char residue analysis shows that PPy-MAPP can promote a generation of more phosphorus-rich structures in the condensed phase that improve the integrity and intumescence of char against fire. The mechanical test indicates that PPy-MAPP has a less negative effect on the tensile strength and elastic modulus of epoxy resin due to the good compatibility between PPy-MAPP and the EP matrix, as supported by differential scanning calorimetry (DSC) analyses. In this paper, these attractive features of PPy-MAPP provide a new strategy to prepare satisfactory flame retardant and super flame retarding EP composites.

## 1. Introduction

Epoxy resin (EP) is a type of thermosetting polymer with a wide application in the fields of the national defense industry and automobile industry [1,2,3]. However, the high fire risk of EP limits its further application, especially in aerospace, electronics and other special fields. Generally, the limiting oxygen index (LOI) value of EP is only about 19.0%. Meanwhile, EP will release many toxic gases during combustion, which is an awful threat to human possessions, life and safety evacuation [4,5,6]. Therefore, many efforts have focused on the study and application of various flame retardants to reduce the fire risk of epoxy resin [7].

The performance of epoxy resin is greatly affected by additives. Different additives have different physicochemical properties, and their effects on the properties of epoxy resin are also different [8,9]. Mineral additives, such as sepiolite, zeolite, pumice, diorite and ocher [10,11,12], can significantly enhance the surface hardness, tensile strength and heat resistance of epoxy resin at high temperatures. In addition to enhancing the tensile strength of epoxy resin, natural fibers, such as sawdust, also play a positive role in enhancing flexural strength [13]. In addition, the particle size and dispersion of additives have a great influence on the physical properties of epoxy resin. Generally, the additive with a small particle size can cause higher physical and mechanical properties in EP [14,15]. Bao et al. [16] used silicon-containing flame retardant (PMDA) to modify carbon nanotubes (CNTs), and the obtained modified CNTs showed better dispersion performance in epoxy resin, showing good flame retardancy and thermal stability. Mostovoy et al. [17] reported the feasibility of using two kinds of phosphates (oligo (resorcinophenyl phosphate) and trichloropropyl phosphate) as epoxy resin plasticizers. The synthesized epoxy resin has excellent bending and impact properties and also has a positive effect on the improvement of flame-retardant properties. Zhou et al. [18] used phosphoric acid modified lignin as a flame-retardant modifier of epoxy resin. The results showed that although epoxy resin showed good flame-retardant properties and thermal stability, it sacrificed the tensile properties and elastic modulus of epoxy resin.

In recent years, ammonium polyphosphate (APP) has received extensive attention because of its high flame-retardant performance, and often acted as an acid source and a gas source in an intumescent system, which made it widely applied in the field of flame-retarded materials. However, APP has high hygroscopicity and poor compatibility with polymer substrate. The combination of APP and polymer organic matter will lead to a serious deterioration of mechanical properties, which limits its application in polymer materials [19,20]. Many efforts have tried to overcome the shortcomings of APP with modification methods. At present, modifications of APP mainly include surfactant modification [7,21], coupling modification [22,23], microencapsulation [24] and melamine modification [25]. Among them, microencapsulation has attracted extensive attention in the field of flame-retarded materials because of its controllable effect, strong variability and excellent compatibility.

Polypyrrole (PPy) is a type of polymer material with good stability and strong conductivity and is widely used in the fields of supercapacitors, lithium electronic batteries, energy storage and heavy metal adsorption [26,27,28]. PPy is mainly composed of a carbon atom and a nitrogen atom, which has potential application as a carbon source in intumescent systems. Pethsangave et al. [29] synthesized the functionalized polypyrrole-loaded graphene nanocomposites (G-fPPy) and then coated them on the surface of cotton fabrics and wood to obtain excellent flame-retardant materials. Wang et al. [30] wrapped copper phytate-doped PPy on the surface of boron nitride nanosheets and found that PPy modified boron nitride could improve both the flame retardancy and mechanical properties of polyurethane (PU) materials. These comprehensive studies indicate that PPy may be an ideal flame-retarded composition in a flame-retardant system.

In this work, PPy-MAPP was synthesized by microencapsulating MAPP with PPy, and the chemical composition, morphological characteristics and thermal stability of PPy-MAPP were characterized by Fourier transform infrared spectroscopy (FTIR), scanning electron microscopy (SEM), and a thermogravimetric analyzer (TG). Then, the obtained PPy-MAPP was added to the EP matrix to prepare flame-retarded EP composites. The flame retardancy, smoke suppression, thermal stability and mechanical properties of EP composites were evaluated by LOI, UL94, TG and cone calorimeter tests. Finally, the possible flame-retardant and smoke suppression mechanisms of PPy-MAPP in the EP matrix were proposed.

## 2. Materials and Methods

### 2.1. Materials

Ammonium polyphosphate (APP, form II) and melamine (MEL) were supplied by Hangzhou JLS Flame Retardants Chemical Co., Ltd. (Hangzhou, China). The phosphorus content of ammonium polyphosphate is 31~32%, the water solubility at 25 °C is less than 0.5, and the particle size is 18 μm. Pyrrole (PY) and ferric chloride were purchased from Shanghai Macklin Biochemical Co., Ltd. (Shanghai, China). The epoxy resin (E-44) was obtained from Zhenjiang Danbao Resin Co., Ltd. (Zhenjiang, China). The epoxy value of the epoxy resin is 0.41~0.47 per 100 g, and the production standard is GB13657-1992. 4,4′-diaminodiphenylmethane (DDM) was purchased from Changzhou Runxiang Chemical Co., Ltd. (Changzhou, China). The molecular weight of DDM is 198.26 g/mol and its melting point is 89~91 °C. Ethanol was supplied by Tianjin Hengxing Chemical Reagent Manufacturing Co., Ltd. (Tianjin, China). All the reagents were used as received.

### 2.2. Synthesis of PPy-MAPP

PPy-MAPP was prepared through microencapsulation using polypyrrole after a melamine coating procession. Firstly, APP (60 g) was dissolved in a 400 mL mixed solution (volume (ethanol): volume (water) = 1:1), and then the above solution was dispersed by ultrasonics at 50 °C for 30 min. Then, MEL (6 g) was introduced into the solution with ultrasonic dispersion continued for 10 min. The above solution was injected into a three-necked flask with magnetic stirring for 1 h at 120 °C. After that, the mixture was adjusted to 90 °C for 2 h stirring. Finally, 100 mL of ferric chloride solution with a mass concentration of 0.9 g L^−1^ was slowly dropped into the above solution with stirring for 12 h. The obtained product was filtered, and then washed with water and ethanol 3 times, and then dried in an oven at 60 °C for 24 h to obtain PPy-MAPP.

### 2.3. Preparation of Pure EP and EP Composites

Firstly, EP was preheated at 60 °C in advance. Then, a certain amount of PPy-MAPP was added to the epoxy resin and stirred at 60 °C for 10 min at a stirring speed of 400 r min^−1^. Afterwards, a certain amount of DDM as a curing agent was introduced into the above epoxy resin mixture for stirring at a speed of 400 r min^−1^ for 5 min. The mixture was poured into a PTFE mold for 2 h at 120 °C and then heated at 150 °C for 1 h. Finally, EP composites were obtained after demolding. The synthesis process of PPy-MAPP and the resulting EP composites is shown in Figure 1. The composition and flammability of the EP composites are shown in Table 1.

### 2.4. Characterization and Measurements

Fourier transform infrared (FTIR) spectra were obtained on an I CAN 9 infrared spectrometer (Tianjin Energy Spectrum Technology Co., Ltd., Tianjin, China) in a wavenumber range of 4000–500 cm^−1^.

Scanning electronic microscopy (SEM) was utilized to obtain the microcosmic morphologies of the samples at an accelerating voltage of 20 kV on a TESCAN MIRA3 LMU (TESCAN CHINA, Ltd., Brno, Czech Republic). Energy dispersive spectroscopy (EDS) was equipped to study the element analysis on an X-Max20 X-ray probe (Oxford Instruments Co., Ltd., Abingdon Oxon, UK).

Thermogravimetric analysis (TGA) was obtained on a TGA/SD-TA851e thermal gravimetric analyzer (Mettler Toledo International Trading Co., Ltd., Shanghai, China). The sample (about 5 mg) was heated from 25 to 800 °C under air atmosphere of 50 mL min^−1^. The theoretical char residue (*W*_cal_) was calculated according to Formula (1):(1)$$W_{\mathrm{cal}}(T)=\sum_{i=1}^{n}\chi_{i}W_{i}(T)$$
where *χ_i_* is the content of compound *i*, and *W_i_* is the experimental weight of compound *i* after the TG test.

Differential scanning calorimetry (DSC) analysis was performed using the DSC823e (Mettler Toledo International Trading Co., Ltd., Shanghai, China). The sample was heated to 200 °C and maintained for 3 min, and then cooled down to room temperature with a cooling rate of 10 °C min^−1^ to eliminate the thermal history. Then, the sample was heated at a heating rate of 10 °C min^−1^ from 25 °C to 200 °C under a nitrogen 50 mL min^−1^ atmosphere.

The limiting oxygen index (LOI) values of pure EP and EP composites were obtained using an HC-2CZ oxygen index meter (Nanjing Shangyuan Analytical Instrument Co. Ltd., Nanjing, China) according to ASTM D2863-19, “Standard test method for measuring the minimum oxygen concentration to support candle-like combustion of plastics (oxygen index)” with a sample size of 130 × 6.5 × 3.2 mm^3^.

The UL94 test was conducted using a JL8333-3 horizontal combustion tester (Nanjing Jionglei Instrument Equipment Co., Ltd., Nanjing, China). The test is in accordance with the procedure of UL94-2016, “Standard for safety: Tests for flammability of plastic materials for parts in devices and appliances”, and the specimens for testing were 130 × 3 × 3.2 mm^3^.

A cone calorimeter was utilized using the FTT iCone classic instrument model FTT0001 (Fire Testing Technology Co., Ltd., East Grinstead, UK) to investigate the combustion behavior of the EP composites according to ISO 5660-1:2015, “Reaction-to-fire tests-Heat release, smoke production and mass loss rate-Part 1: Heat release rate (cone calorimeter method) and smoke production rate (dynamic measurement)”. All the samples for testing were horizontally irradiated at a heat flux of 50 kW m^−2^ with a size of 100 × 100 × 3.5 mm^3^.

The mechanical test was performed in a WDW-10D microcomputer controlled electronic universal testing machine (Jinan Hengsi Shengda Instrument Co., Ltd., Jinan, China) according to GB/T1040.2-2006, “Plastics-Determination of tensile properties-Part 2: Test conditions for molding and extrusion plastics”. The sample size was a type I standard sample and the test speed was 5 mm min^−1^. Each sample was tested three times, and the final experimental results were taken as the average of the three experiments.

## 3. Results and Discussion

### 3.1. Characterization of PPy-MAPP

FTIR spectra of MAPP and PPy-MAPP are shown in Figure 2. PPy-MAPP, MAPP and APP show similar characteristic peaks, which means that the original functional groups of APP have not changed after the modification process. Compared with MAPP, the peaks of 3400, 1625 and 1459 cm^−1^ are different in the spectrum of PPy-MAPP. The characteristic peak of 3400 cm^−1^ is assigned to the stretching vibration of the N–H group, while the peaks at 1625 and 1459 cm^−1^ are assigned to the vibration of the aromatic ring in the polypyrrole structure [31].

Figure 3 shows the morphologies of the APP, MAPP and PPy-MAPP flame retardants. It can be seen that the surface of APP is smooth without impurities, while MAPP is slightly rough, indicating that MEL successfully grafted onto the surface of APP. The surface of PPy-MAPP is rougher and flaky, which means that PPy is successfully wrapped on the surface of MAPP. From the results of EDS, the contents of carbon and nitrogen in PPy-MAPP are higher than that of APP and MAPP, further verifying that PPy-MAPP was synthesized successfully.

The thermal decomposition processes of APP and MAPP and PPy-MAPP were investigated by TG analysis. As shown in Figure 4, both MAPP and PPy-MAPP have two obvious decomposition processes at 250–350 °C and 350–500 °C, respectively. In addition to the obvious thermal decomposition of APP between 250 °C and 500 °C, a new thermal decomposition process appears at 500 °C and 700 °C, and the peak thermal decomposition is about 690 °C. By comparison, MAPP and PPy-MAPP have less thermal decompositions after 500 °C. Compared to MAPP, PPy-MAPP has a less mass loss at the temperature range of 500–800 °C, indicating an improvement in thermal stability at high temperatures. In addition, the residual weights of APP, MAPP and PPy-MAPP at 800 °C are 49.1%, 61.4% and 63.2%, respectively, further indicating a better char-forming ability of PPy-MAPP.

### 3.2. TG Analysis

TG analysis was utilized to evaluate the thermal stability of EP composites, and the TG and DTG curves are shown in Figure 5. From Figure 5, all the samples mainly decompose in the temperature range of 350–500 °C. Compared to pure EP, the mass loss of the EP composites containing either MAPP or PPy-MAPP is obviously decreased, which indicates that MAPP and PPy-MAPP can enhance both the charring ability and thermal stability of EP. Especially, the EP composite containing PPy-MAPP has an earlier decomposition and a lower PMLR value, indicating that PPy-MAPP changes the thermal decomposition process of epoxy resin.

The specific data for the thermal decomposition of EP composites are listed in Table 2. It can be seen that the *T*_5%_ and *T*_max_ values of pure EP are 396.1 °C and 421.4 °C, respectively. After adding PPy-MAPP, the T_5%_ and T_max_ values of EP composites are decreased obviously, among which EP/15PPy-MAPP has the lowest *T*_5%_ and *T*_max_ values, which is due to the low decomposition temperature of PPy-MAPP containing weak bonds of N-H and P-O-C [32]. The char residue of the EP composite containing 15 wt% PPy-MAPP increases to 33.8% from 19.9% of pure EP, exhibiting a super char-forming ability. In order to further assess the char-forming behavior of PPy-MAPP, ΔW can be utilized for quantitative analysis of the char-forming ability [33]. The value of ΔW is positively correlated with the content of PPy-MAPP, which indicates that the char-forming ability of EP composites increases with PPy-MAPP content. Especially, the ΔW value of EP/15PPy-MAPP is obviously improved by 17.7% compared with EP/15MAPP, meaning the PPy-MAPP could enhance the char-forming ability of EP composites. To sum up, the microencapsulation of MAPP by polypyrrole obviously enhances the char-forming ability of flame-retarded EPs.

### 3.3. DSC Analysis

The glass transition temperature (*T*_g_) of EP composites is measured by DSC analysis, and the results are presented in Figure 6. As shown in Figure 6, the *T*_g_ value of pure EP is 170 °C. The *T*_g_ value of EP composites decreases gradually with the addition of flame retardant, which indicates that the presence of either MAPP or PPy-MAPP will directly affect the cross-linking network of the EP and reduce the cross-linking degree of the EP [34]. On the other hand, the addition of PPy-MAPP has a less negative effect on the glass transition temperature of epoxy resin compared with MAPP, indicating the better compatibility between PPy-MAPP and epoxy resin.

### 3.4. LOI and UL94 Tests

The flame retardancy of EP composites was evaluated using the LOI and UL94 tests, and the corresponding results are listed in Table 1 and Figure 7. As shown in Table 1, pure EP has high flammability with an LOI value of 26.3% and fails to pass the UL94 test. The increased degree of the LOI value and UL94 rating of EP composites vary with the content of flame retardants. Both MAPP and PPy-MAPP impart good flame-retardant performance to the EPs, among which EP/15PPy-MAPP has the best flame-retardant effect with an LOI value of 42.4% and passes the UL94 V-0 rating. It is noteworthy that the same content of PPy-MAPP imparts a higher LOI value to EP composite compared to MAPP, indicating that PPy-MAPP has a higher flame-retardant efficiency than that of MAPP.

### 3.5. Cone Calorimeter Test

In order to assess the fire risk of EP composites, the flammability of EP composites was tested by a cone calorimeter. Various cone calorimeter parameters, such as the time to ignition (TTI), total heat release (THR), peak heat release rate (PHRR), total smoke release (TSR), total smoke production (TSP) and peak smoke production rate (PSPR) are shown in Table 3 and Figure 8.

From Table 3, it is easily found that the TTI value of EP/15PPy-MAPP shows a slight increase compared with pure EP, while EP/15MAPP has no obvious change. From Figure 8a, pure EP burns violently after ignition and is concomitant with the release of a large amount of combustible gases, with its value of THR and PHRR being 107.9 MJ/m^2^ and 1138.2 KW/m^2^, respectively. Compared with pure EP, the PHRR value of EP/15PPy-MAPP and EP/15MAPP is 430.7 and 570.9 KW/m^2^, respectively, which is decreased by 62.2% and 49.8%, respectively. From Figure 8b, the THR value of EP/15PPy-MAPP is reduced by 58.0% while EP/15MAPP is only reduced by 36.7% compared to pure EP, further indicating that PPy-MAPP has a higher flame-retardant efficiency.

The major factor leading to death is smoke in most cases of real fire scenarios; therefore, it is necessary to assess the parameters of smoke release. The smoke production rate (SPR) and total smoke production (TSP) curves of the EP composites are shown in Figure 8c,d. The SPR and TSP values of EP decrease obviously after the addition of PPy-MAPP. Compared with pure EP, the PSPR and TSP values of EP/15PPy-MAPP decrease by 62.2% and 73.9%, respectively, while the corresponding data of EP/15MAPP is reduced by 51.6% and 55.2%, respectively. It indicates that PPy-MAPP has a more satisfactory smoke suppression efficiency in the EP matrix than MAPP.

Figure 9 shows the digital photos of the char residues of the pure EP and EP composites after the cone calorimeter test. In Figure 9a, pure EP has a char height of 17 mm, concomitant with a fragmented char after combustion. After the addition of PPy-MAPP and MAPP, the char residue of EP/15PPy-MAPP and EP/15MAPP has more integrity and expansion compared to pure EP, as seen in Figure 9b,c. The char height of EP/15PPy-MAPP is 78 mm, while EP/15MAPP is 56 mm, indicating a better char-forming ability of PPy-MAPP than MAPP. From Figure 9b*,c*, EP/15PPy-MAPP shows a more compact intumescent char with fewer voids and holes compared with EP/15MAPP, meaning the same addition of PPy-MAPP is beneficial to form a more integrated and intumescent char against fire. In order to further analyze our research results, we compare the performance of PPy-MAPP with other papers, and the details are shown in Table 4.

### 3.6. Char Residue Analysis

The SEM images and EDS maps of the char residues for the EP composites after the cone calorimeter test are shown in Figure 10. In Figure 10a, it can be observed that the surface of EP/15MAPP char has a lot of large pore structures, while the surface of EP/15PPy-MAPP char has a denser structure with smaller pores. It indicates that PPy-MAPP can effectively enhance the integrity and compactness of the char layer during combustion. From the EDS maps of char residues, the EP/15PPy-MAPP char has a higher phosphorus content compared to the EP/15MAPP char, which means more phosphorus-containing cross-linked structures are formed. The mass ratio of carbon and oxygen reflects the thermal stability and oxidation degree of the char residues, and a higher C/O ratio presents a better heat insulation property [36,37]. The C/O ratio of the EP/15PPy-MAPP char is higher than that of the EP/15MAPP char, indicating that EP/15PPy-MAPP generates a more intumescent and compact char with a better heat insulation effect.

To investigate the flame-retardant mechanism of PPy-MAPP, FTIR spectra of EP/15MAPP and EP/15PPy-MAPP treated at different temperatures are shown in Figure 11. It can be observed that both EP/15MAPP and EP/15PPy-MAPP show the obvious absorption peaks of the C=C group at 1600 cm^−1^, the C-C group in the aromatic ring at 1450 cm^−1^ and the P-O-C group at 1089 cm^−1^ [38,39]. When the temperature is below 450 °C, EP/15MAPP and EP/15PPy-MAPP have little difference in the types and intensities of chemical groups. When the temperature increases above 450 °C, the intensities of the peaks assigned to P-O-C and C=C groups in the spectrum of the EP/15PPy-MAPP sample are obviously stronger than those of EP/15MAPP, indicating that the addition of PPy-MAPP generates more aromatic and cross-linked structures in the condensed phase that effectively enhances the barrier effect and thermal stability of char residue. The possible flame-retardant mechanism of EP/PPy-MAPP composites is proposed in Figure 12.

### 3.7. Tensile Test

The results of the tensile tests of the pure EP and EP composites are shown in Table 5. From Table 5, the tensile strength of the samples shows a decreasing trend with the increasing content of flame retardant, which is due to the addition of fillers that easily causes concentrated stress. In addition, EP is a polymer compound formed by the interaction of covalent bonds and intermolecular forces, which has poor compatibility with inorganic ammonium polyphosphate bound by ionic bonds. The surface of PPy-MAPP has fewer ionic bonds due to the microencapsulation of PPy compared to MAPP [40], which indicates that PPy-MAPP has a less negative effect on the cross-linking process of EP. The elastic modulus of EP composites has been enhanced by adding PPy-MAPP, which may be due to the reaction between the N–H bond on the surface of polypyrrole and the cross-linking network of epoxy resin.

## 4. Conclusions

In this study, a mono-component intumescent flame retardant, named PPy-MAPP, was successfully prepared and then applied in epoxy resin to reduce its fire hazard. The flammability, thermal stability and mechanical properties of the EP composites were carefully characterized. The results show that PPy-MAPP exerts better flame-retardant and smoke suppression efficiencies in the EP matrix compared to the same addition of melamine-coated ammonium polyphosphate. In particular, EP/15PPy-MAPP can reach an LOI value of 42.4% and easily achieve a UL94 V-0 rating, accompanied by a reduction of 61.9% in the THR value and 65.6% in the PHRR value compared to pure EP. Moreover, the PSPR and TSP values of EP/15PPy-MAPP are 62.2% and 73.9%, respectively, lower than those of pure EP. The char residue analysis shows that PPy-MAPP can promote the generation of phosphorus-rich cross-linked and aromatic structures in the condensed phase that improves the integrity and intumescence of char residue against heat and mass transfer. The tensile test shows that Ppy-MAPP increases the elastic modulus of EP composites concomitant with a less negative effect on the tensile strength due to the good compatibility between the EP matrix and PPy-MAPP, as supported by the DSC analysis. Furthermore, the elastic modulus of EP/15PPy-MAPP is 38.5% higher than that of pure EP. Overall, these attractive features of PPy-MAPP provide a new insight to obtain EP composites with satisfactory flame retardancy, smoke suppression and mechanical properties. 

## Figures and Tables

**Figure 1 polymers-14-02375-f001:**
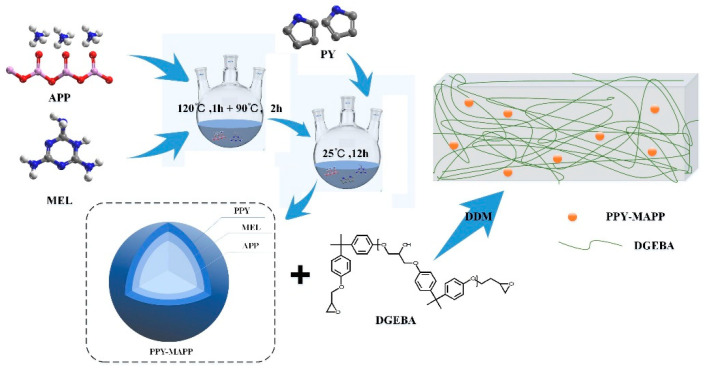
Preparation process of PPy-MAPP and EP composites.

**Figure 2 polymers-14-02375-f002:**
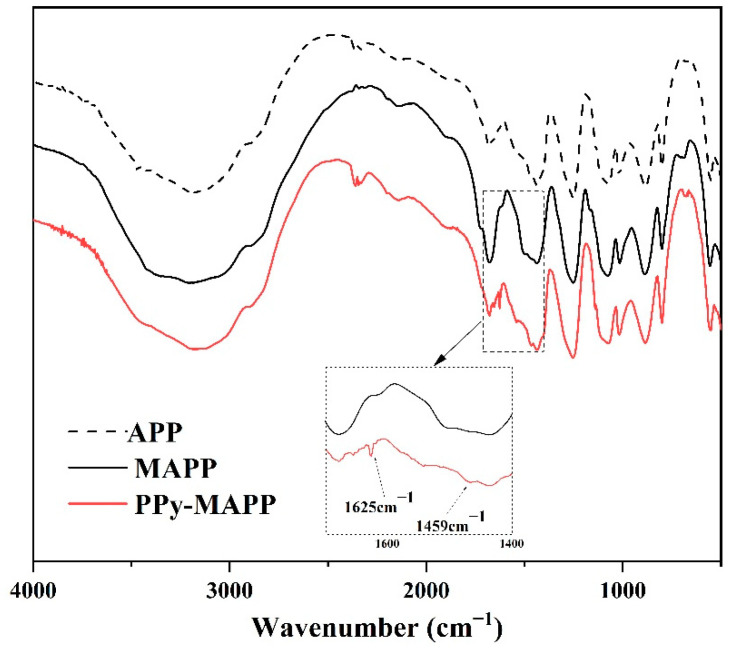
FTIR spectra of APP, MAPP and PPy-MAPP.

**Figure 3 polymers-14-02375-f003:**
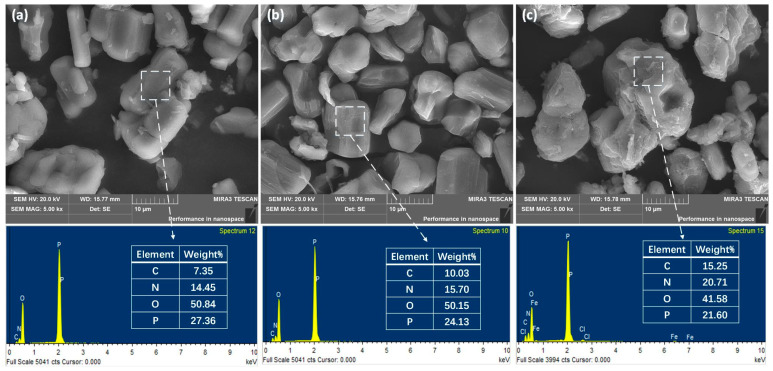
Microstructure of APP (**a**), MAPP (**b**) and PPy-MAPP (**c**).

**Figure 4 polymers-14-02375-f004:**
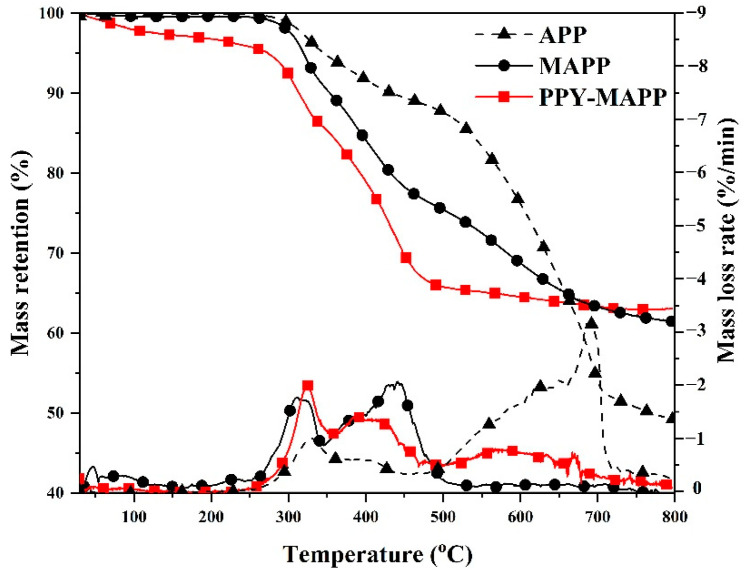
TG and DTG curves of APP and MAPP and PPy-MAPP.

**Figure 5 polymers-14-02375-f005:**
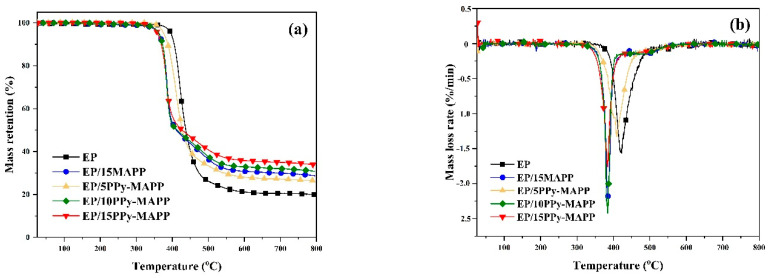
TG (**a**) and DTG (**b**) curves of EP and flame-retarded EPs.

**Figure 6 polymers-14-02375-f006:**
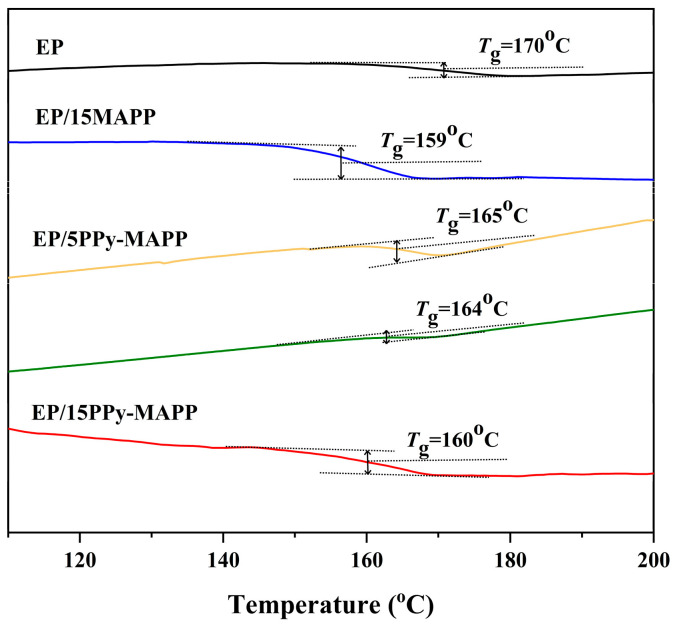
DSC curves of flame-retardant epoxy resin.

**Figure 7 polymers-14-02375-f007:**
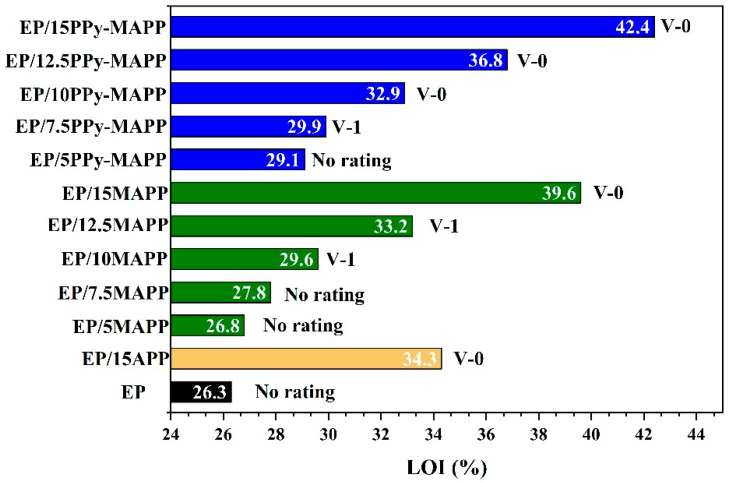
LOI value and UL94 rating of neat EP and EP composites.

**Figure 8 polymers-14-02375-f008:**
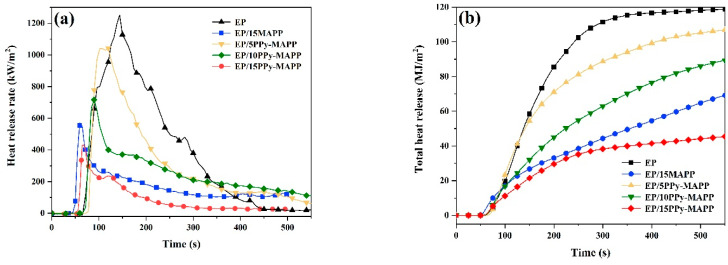
Cone calorimeter data of epoxy composites: heat release rate (**a**), total heat release (**b**), smoke production rate (**c**) and total smoke production (**d**).

**Figure 9 polymers-14-02375-f009:**
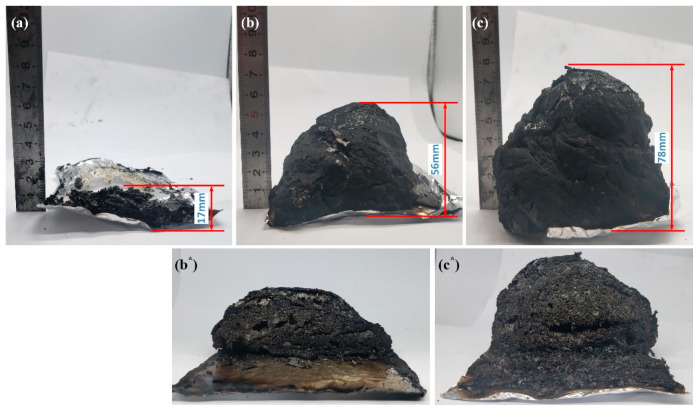
Char morphologies of EP (**a**), EP/15MAPP (**b**), EP/15PPy-MAPP (**c**), profile map of EP/15MAPP (**b***) and EP/15PPy-MAPP (**c***).

**Figure 10 polymers-14-02375-f010:**
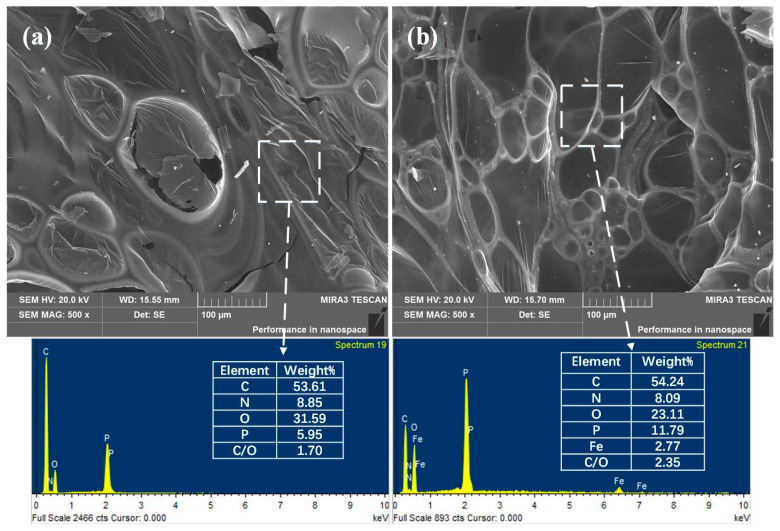
SEM image of EP/15MAPP (**a**) and EP/15PPy-MAPP (**b**) after cone calorimetry test.

**Figure 11 polymers-14-02375-f011:**
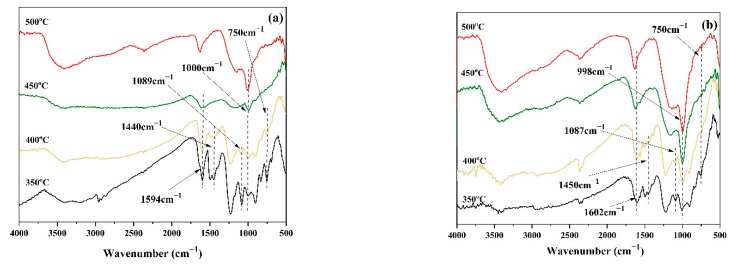
FTIR spectra of EP/15MAPP (**a**) and EP/15PPy-MAPP (**b**) under different heating temperatures.

**Figure 12 polymers-14-02375-f012:**
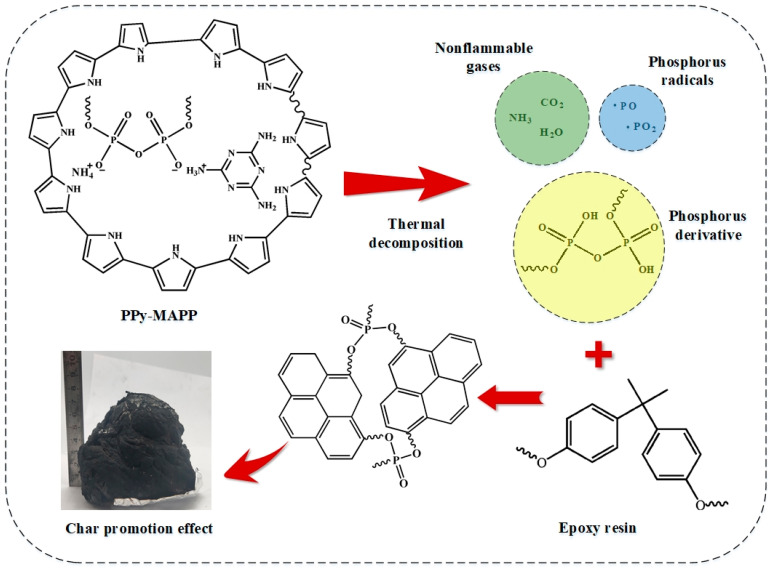
Flame-retardant mechanism of EP/PPy-MAPP composites.

**Table 1 polymers-14-02375-t001:** Formulations and flammability of pure EP and EP composites.

Samples	APP (wt%)	MAPP (wt%)	PPy-MAPP (wt%)	EP (wt%)	DDM (wt%)
EP	0	0	0	80	20
EP/15APP	15	0	0	68	17
EP/5MAPP	0	5	0	76	19
EP/7.5MAPP	0	7.5	0	74	18.5
EP/10MAPP	0	10	0	72	18
EP/12.5MAPP	0	12.5	0	70	17.5
EP/15MAPP	0	15	0	68	17
EP/5PPy-MAPP	0	0	5	76	19
EP/7.5PPy-MAPP	0	0	7.5	74	18.5
EP/10PPy-MAPP	0	0	10	72	18
EP/12.5PPy-MAPP	0	0	12.5	70	17.5
EP/15PPy-MAPP	0	0	15	68	17

**Table 2 polymers-14-02375-t002:** Typical thermal data of flame-retarded epoxy resins.

Samples	*T*_5%_ (°C)	*T*_max_ (°C)	PMLR (% min^−1^)	*W*_exp_ (%)	*W* _cal_	Δ*W*
EP	396.1	421.4	1.56	19.9	–	–
EP/15MAPP	362.4	384.5	2.19	28.7	26.15	2.55
EP/5PPY-MAPP	376.5	408.5	1.33	26.4	21.95	4.45
EP/10 PPy-MAPP	363.3	384.5	2.42	30.8	24.30	6.50
EP/15 PPy-MAPP	359.5	384.2	1.76	33.8	26.45	7.35

Note: *T*_5%_—initial decomposition temperature, *T*_max_—the temperature of peak mass loss rate, PMLR—peak value of mass loss rate, Δ*W* = *W*_exp_ − *W*_cal._

**Table 3 polymers-14-02375-t003:** Cone calorimeter data of pure EP and the flame-retarded EP composites.

Samples	TTI (s)	THR (MJ/m^2^)	PHRR (KW/m^2^)	TSR (m^2^/m^2^)	TSP (m^2^)	PSPR (m^2^/s)	Residue (%)
EP	58	118.8	1252.0	1186.7	32.6	0.304	6.8
EP/15MAPP	59	68.3	570.9	866.1	14.6	0.147	24.7
EP/5PPy-MAPP	59	106.1	1040.0	826.9	25.9	0.205	16.9
EP/10PPy-MAPP	61	89.0	896.7	753.4	23.0	0.161	17.1
EP/15PPy-MAPP	67	45.3	430.7	702.4	8.5	0.115	30.3

**Table 4 polymers-14-02375-t004:** Comparison of flame retardant property and smoke suppression performance between EP/15PPy-MAPP and other flame retardant additives in epoxy resin.

Samples	Content (wt%)	LOI (%)	UL94 Rating	Reduction of PHRR (%)	Reduction of PSPR (%)	Char Residue Weight (%)	Refs.
EP/15PPy-MAPP	15	42.4	V-0	65.6	62.2	30.3	this work
EP/DDS/PBI-1.0	14.3	33.5	V-0	49.5	–	21.2	[2]
EP/APP/Fe-3	9.5	30.0	V-0	75.8	–	23.5	[20]
EP/15DDP	15	37.1	V-0	40.8	47.8	34.0	[32]
EP/12.5% DOPO-ABZ	12.5	32.0	V-0	43.0	–	20.1	[34]
EPMAPP12	12	32.5	V-0	84.6	–	–	[35]

Annotation: Content is defined as the addition of flame retardant additives; the calculation of Reduction of PHRR and PSPR are based on pure epoxy resin; EP/DDS/PBI-1.0, DGEBA/dapsone/ is a flame retardant composed of phosphaphenanthrene, benzothiazole and imidazole groups (PBI); EP/APP/Fe-3, DEGBA/DDM/2.0 g β-FeOOH nanorods and 7.5 g APP; EP/15DDP, DEGBA/DDM/ DDM modified APP(DDP); EP/12.5% DOPO-ABZ, DEGBA/DDM/ is the combination of DOPO and 2-aminobenzothiazole (DOPO-ABZ); EPMAPP12, DEGBA/ethylenediamine/a is encapsulated APP using tetraethyl orthosilicate and octyltriethoxysilane (MAPP).

**Table 5 polymers-14-02375-t005:** Mechanical property parameter of pure EP and EP composites.

Samples	Elastic Modulus (MPa)	Tensile Strength (MPa)
EP	5710.3 ± 10.2	51.2 ± 2.4
EP/15MAPP	7556.8 ± 23.1	33.2 ± 1.9
EP/5PPy-MAPP	5792.7 ± 16.4	46.0 ± 2.3
EP/10PPy-MAPP	5814.8 ± 8.9	39.2 ± 2.2
EP/15PPy-MAPP	7907.0 ± 22.0	34.0 ± 1.6

## Data Availability

The data presented in this study are available upon request from the corresponding author.

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
