# Peer review of "Facile Construction of Polypyrrole Microencapsulated Melamine-Coated Ammonium Polyphosphate to Simultaneously Reduce Flammability and Smoke Release of Epoxy Resin"

_polymers, 2022, doi:10.3390/polym14122375_

Round 1
Reviewer 1 Report
The manuscript under the title: “Facile construction of polypyrrole microencapsulated melamine-coated ammonium polyphosphate to simultaneously reduce flammability and smoke release of epoxy resin” is in line with Polymers journal. This topic is relevant and will be of interest to the readers of the journal. It based on original research. This research has scientific novelty and practical significance. The article has a typical organization for research articles.
Before the publication it requires significant improvements, especially:
1. The "Introduction" section: it has been proven that the effect of various modifying additives and fillers on the flammability reduction and physical and mechanical properties of epoxy polymer composites is determined by many factors: ……. I think the related references should be cited corresponding to each aspect, e.g. (but not limited to these), which will undoubtedly improve the "Introduction" section:
- Polymers 2020, 12(7), 1437; https://doi.org/10.3390/polym12071437
- Polymers 2021, 13(19), 3332; https://doi.org/10.3390/polym13193332
- Mater. Appl. Res. 2019, 10, 1135–1139, https://doi.org/10.1134/S2075113319050228
- Polymer Composites. 2020; 41: 2025–2035. https://doi.org/10.1002/pc.25517
- Polymers 2021, 13(15), 2421; https://doi.org/10.3390/polym13152421
2. Section 2.1. It is necessary to add the physicochemical characteristics of all components - give a table with the main physicochemical and technological properties of epoxy resin, hardener and APP.
3. It is necessary to add data on the change in the viscosity of the epoxy composition with the introduction of fillers.
4. Fig.2. Add curve for original ammonium polyphosphate.
5. Fig.4. Add and discuss curves for the original ammonium polyphosphate.
6. Table 2. Add and discuss data for composites containing the original ammonium polyphosphate.
7. Table 3 and Fig. 8. Add and discuss data for composites containing the original ammonium polyphosphate.
8. Fig.9. Add and discuss data for composites containing the original ammonium polyphosphate.
9. Table 3. Add and discuss data for composites containing the original ammonium polyphosphate.
10. Discussion: please compare achieved results with up-to-date literature, also with composites with other admixtures. Discuss the achieved results.
Author Response
Dear reviewer,
I would like to thank the you for your constructive comments on our manuscript “Facile construction of polypyrrole microencapsulated mela-mine-coated ammonium polyphosphate to simultaneously re-duce flammability and smoke release of epoxy resin” (polymers-1745553). I have considered the comments very carefully and have revised the paper accordingly. All changes to the text and figures are shown. I believe that this revised paper has been improved considerably. I hope that the corrections are satisfactory.
According to your comments, we have mainly made some modifications, as follows:
- The "Introduction" section: it has been proven that the effect of various modifying additives and fillers on the flammability reduction and physical and mechanical properties of epoxy polymer composites is determined by many factors: ……. I think the related references should be cited corresponding to each aspect, e.g. (but not limited to these), which will undoubtedly improve the "Introduction" section:
Polymers 2020, 12(7), 1437; https://doi.org/10.3390/polym12071437
Polymers 2021, 13(19), 3332; https://doi.org/10.3390/polym13193332
Mater. Appl. Res. 2019, 10, 1135–1139, https://doi.org/10.1134/S2075113319050228
Polymer Composites. 2020; 41: 2025–2035.
Polymers 2021, 13(15), 2421; https://doi.org/10.3390/polym13152421
Based on the references provided, relevant content has been added to the introduction section.
- Section 2.1. It is necessary to add the physicochemical characteristics of all components - give a table with the main physicochemical and technological properties of epoxy resin, hardener and APP.
Detailed parameters of ammonium polyphosphate, curing agent and epoxy resin have been added in the article.
- It is necessary to add data on the change in the viscosity of the epoxy composition with the introduction of fillers.
We believe that this paper aims to study the flame retardant properties of epoxy resin after curing, so we will record the viscosity changes of epoxy resin before curing in subsequent studies.
- 2. Add curve for original ammonium polyphosphate.
The original APP infrared spectrum curve has been added in Figure 2 of the new manuscript.
- 4. Add and discuss curves for the original ammonium polyphosphate.
Thermogravimetric curve of pure APP has been added in Figure 4 and related descriptions have been added.
- Table 2. Add and discuss data for composites containing the original ammonium polyphosphate
- Table 3 and Fig. 8. Add and discuss data for composites containing the original ammonium polyphosphate.
- 9. Add and discuss data for composites containing the original ammonium polyphosphate.
- Table 3. Add and discuss data for composites containing the original ammonium polyphosphate
For the original APP, we have shown the flame retardant effect of the original APP through the limiting oxygen index and UL94 grade in the article.
- Discussion: please compare achieved results with up-to-date literature, also with composites with other admixtures. Discuss the achieved results.
A comparison with other research results has been added at the end of the article.
Note: In the response letter, all the responses are shown in blue, and all the reviewer’s comments/suggestions are shown in black. Also, in the revised paper, all changes and additions are highlighted in blue.
Best wishes
Liao Jiahao

Reviewer 2 Report
The manuscript entitled "Facile construction of polypyrrole microencapsulated melamine-coated ammonium polyphosphate to simultaneously reduce flammability and smoke release of epoxy resin" has been reviewed. The results are helpful. However, the manuscript needs to be well revised before acceptance. Detailed comments are as follows:
- The quality of English and writing style should be better.
- The full names of some abbreviations, such as DSC, TG, MAPP and PU should be given where the full names and abbreviations first appeared in both Abstract and main text.
- Some abbreviations are not proper. For e.g., The abbreviation of thermogravimetric analysis is TGA. TG is the abbreviation of thermogravimetry.
- There are some typo errors in the manuscript. Units should be separated from the numerical value by a space. In tables and figures, word should be separated from the bracket by a space. Degree C should be input in English. Some numbers, e.g., -1 in min-1, should in the subscript form. Please recheck your manuscript.
- Some scientific terminology is wrong. For e.g., elasticity modulus should be elastic modulus. Diaminodi-phenylmethane should be 4,4'-diaminodiphenylmethane.
- The citation with authors in the main text, such as Pethsangave in Line 53, should be in the form of surname et al. or surname and coworkers.
- Detailed information on epoxy resin (E-44), also called DGEBA, should be provided.
- In Fig. 1, PPy-MAPP should be pointed to a microcapsule in the flask. Furthermore, the chemical structure of EP is DGEBA. More importantly, the curing agent, DDM is missing in the scheme.
- In Table 1, (mass fraction) % should be removed. In addition, the unit of raw material should be added.
- Correct unnecessary capitalization of first letters of some phrases, such as Energy Dispersive Spectroscopy.
- Please provide the manufacture of cone calorimeter.
- In 2.4, the number of the sample for tensile tests should be included.
- In Figs. 2 and 11, the y-axis name and unit are missing. In addition, the unit of y-axis should be a. u. (arbitrary unit) since the spectra were vertically translated.
- In Figs. 4 and 5, Mass loss and Mass loss rate should be Mass and Derivative mass or DTG, respectively.
- In Fig. 6, Heat Flow should be Heat flow. The unit should be a.u. (arbitrary unit) since the thermograms had been vertically translated.
- Please remove the LOI and UL 94 results from Table 1, which are the same as Fig. 7.
- In Table 4, the standard deviations are missing.
- References should be well revised under the guide for authors. Some pages or articles numbers are missing. The abbreviations of journal names are not unified. Correct the unnecessary capitalization of first letters of some paper titles.
Author Response
Dear reviewer,
We are submitting a revised version of our manuscript “Facile construction of polypyrrole microencapsulated melamine-coated ammonium polyphosphate to simultaneously reduce flammability and smoke release of epoxy resin” (polymers-1 745553). We would like to thank you for your careful and thoughtful comments on previous draft. We have carefully taken your comment into consideration in preparing our revision, and the revision was highlighted in yellow in the draft.
Our response to your comment is shown below.
- The quality of English and writing style should be better.
In the new manuscript, we have improved the writing style in the article.
- The full names of some abbreviations, such as DSC, TG, MAPP and PU should be given where the full names and abbreviations first appeared in both Abstract and main text.
The abbreviation mentioned has been modified accordingly in the article.
- Some abbreviations are not proper. For e.g., The abbreviation of thermogravimetric analysis is TGA. TG is the abbreviation of thermogravimetry.
The abbreviation TG in the article has been comprehensively checked and modified.
- There are some typo errors in the manuscript. Units should be separated from the numerical value by a space. In tables and figures, word should be separated from the bracket by a space. Degree C should be input in English. Some numbers, e.g., -1 in min-1, should in the subscript form. Please recheck your manuscript.
The units and subscripts in the article have been modified.
- Some scientific terminology is wrong. For e.g., elasticity modulus should be elastic modulus. Diaminodi-phenylmethane should be 4,4'-diaminodiphenylmethane.
Relevant content in the article has been modified.
- The citation with authors in the main text, such as Pethsangave in Line 53, should be in the form of surname et al. or surname and coworkers.
Relevant content in the article has been modified.
- Detailed information on epoxy resin (E-44), also called DGEBA, should be provided.
Details of epoxy resin have been added to the raw material section.
- In Fig. 1, PPy-MAPP should be pointed to a microcapsule in the flask. Furthermore, the chemical structure of EP is DGEBA. More importantly, the curing agent, DDM is missing in the scheme.
Figure 1 has been modified.
- In Table 1, (mass fraction) % should be removed. In addition, the unit of raw material should be added.
Remove relevant content and add raw material units.
- Correct unnecessary capitalization of first letters of some phrases, such as Energy Dispersive Spectroscopy.
Relevant content has been modified.
- Please provide the manufacture of cone calorimeter.
Relevant information of cone calorimeter has been added in 2.4.
- In 2.4, the number of the sample for tensile tests should be included.
Number of tests added at relevant locations.
- In Figs. 2 and 11, the y-axis name and unit are missing. In addition, the unit of y-axis should be a. u. (arbitrary unit) since the spectra were vertically translated.
The ordinate has been added in Fig. 2 and Fig. 11.
- In Figs. 4 and 5, Mass loss and Mass loss rate should be Mass and Derivative mass or DTG, respectively.
Figure 4 and Figure 5 have been modified.
- In Fig. 6, Heat Flow should be Heat flow. The unit should be a.u. (arbitrary unit) since the thermograms had been vertically translated.
Figure 6 has been modified.
- Please remove the LOI and UL 94 results from Table 1, which are the same as Fig. 7.
In the new draft, the relevant content in table 1 was deleted and the specific value of the limiting oxygen index was added in Figure 7.
- In Table 4, the standard deviations are missing.
Related content has been added to the table 4.
- References should be well revised under the guide for authors. Some pages or articles numbers are missing. The abbreviations of journal names are not unified. Correct the unnecessary capitalization of first letters of some paper titles.
References have been modified.
Note: In the response letter, all the responses are shown in blue, and all the reviewer’s comments/suggestions are shown in black. Also, in the revised paper, all changes and additions are highlighted in blue.
Best wishes.
Liao Jiahao

Round 2
Reviewer 1 Report
The authors considered most of the comments or adequately responded to the remarks contained in the review; therefore, the work may be approved for publication.
Author Response
Dear reviewer,
I would like to thank you for your constructive comments on our manuscript “Facile construction of polypyrrole microencapsulated mela-mine-coated ammonium polyphosphate to simultaneously reduce flammability and smoke release of epoxy resin” (polymers-1745553). We have revised the article according to the comments of other reviewers. All changes to the text and figures are shown. Thank you for your valuable comments on our paper.
Best wishes
Liao Jiahao
Point-by-point response
Reviewer Comments:
The authors considered most of the comments or adequately responded to the remarks contained in the review; therefore, the work may be approved for publication.
AN: Thanks you very much.

Reviewer 2 Report
Although some revisions have been done, the manuscript still needs to be well improved before acceptance. Detailed comments are as follows:
1. The full name of TGA is thermogravimetric analysis, not thermogravimetric analyzer and thermogravimetric (TG) analysis (Line 136).
2. In Tables 1-3, the left-half bracket should be separated from the word by a space.
3. In Fig. 6, the left-half bracket should be separated from the word by a space. In addition, the endothermic or exothermic direction should be added.
4. In Fig. 8, correct unnecessary capitalization of first letters of some phrases, such as Total Heat Release.
5. Uncorrected FTIR spectra should be removed from Fig. 11.
Author Response
Dear reviewer,
On behalf of my co-authors, we thank you very much for your positive and constructive comments and suggestions on our manuscript entitled “Facile construction of polypyrrole microencapsulated melamine-coated ammonium polyphosphate to simultaneously reduce flammability and smoke release of epoxy resin” (polymers-1 745553). We have revised the article according to the latest comments.
In this revised version, we have addressed the concerns of your comment. We hope that these revision successfully address your concerns and requirements and that this manuscript will be accepted. Looking forward to hearing from you soon.
Best wishes.
Liao jiahao
Point-by-point response
Reviewer Comments:
According to your comments, we have mainly made some modifications, as follows:
- The full name of TGA is thermogravimetric analysis, not thermogravimetric analyzer and thermogravimetric (TG) analysis (Line 136).
AN: The relevant content in the article has been modified.
- In Tables 1-3, the left-half bracket should be separated from the word by a space.
AN: The relevant content in the article has been modified.
- In Fig. 6, the left-half bracket should be separated from the word by a space. In addition, the endothermic or exothermic direction should be added.
AN: The relevant content in the article has been modified.
- In Fig. 8, correct unnecessary capitalization of first letters of some phrases, such as Total Heat Release.
AN: The relevant content in the article has been modified.
- Uncorrected FTIR spectra should be removed from Fig. 11.
AN: The relevant content in the article has been removed.
Note: In the response letter, all the responses are shown in blue, and all the reviewer’s comments/suggestions are shown in black. Also, in the revised paper, all changes and additions are highlighted in blue.
